# Proactive slip control
# by learned slip model and trajectory adaptation

**Kiyanoush Nazari**
School of Computer Science
University of Lincoln United Kingdom
`19749684@students.lincoln.ac.uk`

**Willow Mandil**
School of Computer Science
University of Lincoln United Kingdom
`18710370@students.lincoln.ac.uk`

**Amir Ghalamzan E.**
Lincoln Institute for Agri-Food Technology
University of Lincoln United Kingdom
`aghalamzanesfahani@lincoln.ac.uk`

**Abstract:** This paper presents a novel control approach to dealing with object slip during robotic manipulative movements. Slip is a major cause of failure in many robotic grasping and manipulation tasks. Existing works increase grip force to avoid/control slip. However, this may not be feasible when (i) the robot cannot increase the gripping force– the max gripping force is already applied or (ii) increased force damages the grasped object, such as soft fruit. Moreover, the robot fixes the gripping force when it forms a stable grasp on the surface of an object, and changing the gripping force during real-time manipulation may not be an effective control policy. We propose a novel control approach to slip avoidance including a learned action-conditioned slip predictor and a constrained optimiser avoiding a predicted slip given a desired robot action. We show the effectiveness of the proposed trajectory adaptation method with the receding horizon controller with a series of real-robot test cases. Our experimental results show our proposed data-driven predictive controller can control slip for objects unseen in training.

(Video and Github)

**Keywords:** Slip avoidance, Robot Control, Manipulation

## 1   Introduction

Sense of touch in humans and primates is key to building sophisticated dexterous manipulation skills [1]. It provides localised contact information on human skin in particular as an effective perception in our daily lives [2]. A slip event occurs when the ratio of applied shear to the normal force on the contact area exceeds the static friction coefficient. Recent studies show that human reflexes to the feeling of slip (we use the term slip to refer to the slipping action of an object between fingers) during manipulative movements not only include grip adjustment but also specific movements of the shoulder and elbow [3]. Existing slip avoidance control methods in robotic manipulation using artificial tactile sensing are limited to: (i) grip force adaptation in case of slip [4, 5], (ii) reactive algorithms that activate after slip onset and detection [6, 7], and (iii) gentle manipulation tasks, *e.g.*, slow lifting movements for 10-20 [cm] [5, 6]. However, increased grip force may not be an effective action when (i) the gripper has already reached its force limits, (ii) the grasped object is delicate/fragile such as soft foods (*i.e.* strawberry where the average grip force to damage the fruit is 1.5 N [8]). Moreover, many grippers, such as Robotiq's Hand-E Gripper or Franka Emika Gripper, do not have enough force resolution, or (iii) the gripper cannot be controlled in real-time due to connection limitations *i.e.* blocking TCP/IP connections in the Franka Emika gripper.

We propose two novel means of controlling slip by adapting the trajectory of manipulative movements: (i) A Reactive Slip Control (**RSC**) policy, which uses detected slip – slip classifier at time $t$– to reduce the robot velocity to avoid slip; (ii) A Proactive Slip Control (**PSC**) policy for avoiding slip

6th Conference on Robot Learning (CoRL 2022), Auckland, New Zealand.

during robotic manipulative movements (see schematic of the approach in Fig. 1). PSC includes an action-conditioned slip predictor, a slip classifier at $t + h$ where $h$ is the prediction horizon using an LSTM (Long-Short Term Memory) based neural network and a predictive controller. The controller minimises the likelihood of predicted slip in a prediction horizon by adjusting the robot's planned movements and deviating from the reference trajectories. We use a quadratic term penalising deviation from a reference trajectory; hence, PSC avoids slip while closely following a given reference trajectory. Our constrained optimisation enables the robot to satisfy its constraints in generating optimal future robot actions. We study the effectiveness of our on-the-fly adaptation of a real robot's pre-planned trajectory (which may be minimum time or minimum jerk) in a series of real robot experiments. We compare *RSC* and *PSC* in terms of their performance in slip avoidance, computation complexity, and optimality. Our experimental results indicate PSC outperforms RSC and can change the game in robotic manipulation success where increased gripping force is not possible.

## 2  Related works

Slip detection algorithms are thoroughly reviewed in [9, 10]. [11] use Support Vector Machine (SVM) and Random Forest (RF) on SynTouch BioTac's raw tactile readings and [12] use SVM on TacTip signals for slip classification. SVM and RF have computational complexity problems in training for large data sets with high feature dimensionality. Specifying a threshold on the rate of shear forces for slip detection is a common approach for slip classification [13, 4]. However, it needs initial friction estimation, and finding the threshold for large object sets can be expensive. Other techniques rely on analysing micro-vibration in the incipient slip phase such as spectral analysis [14, 15]. These approaches can fail when robot vibrations interfere with sensor-object vibrations. Multi-modal approaches are also proposed to use proximity [16] or visual sensing data [17] to improve slip detection. Recurrent Neural Networks and Graph Neural Networks are also used for slip classification [18, 19]. While [18, 19] only use tactile readings for slip detection, we use an LSTM-based model with combined tactile data and robot actions to predict slip in a prediction horizon.

Grip force controllers based on tactile readings are usually designed to avoid slip [11]. [20, 4, 7] increase the normal force until the ratio of normal to shear force becomes smaller than the known/estimated static friction coefficient. This method can lead to deformation or damage of delicate objects, e.g., soft fruits. [4] propose a heuristic to limit deformations of a deformable object but it needs to measure the initial grasp width and object deformation. [21] suggest increasing the number of gripping fingers to avoid slip by using a multi-finger hand. In bi-manual manipulation, [6] performs hand pose correction to bring the resultant force inside the friction cone as a slip avoidance controller. The above approaches have certain limitations where increased grip force is not an option (e.g. (1) Franka Emika gripper is not force-controlled or (2) in delicate object manipulation).

Learning from demonstrations [22, 23] or optimisation can be used for planning robot movements. E.g., online trajectory re-planning is effectively used in areas such as agile quadrotor control [24, 25], but to the authors' knowledge, it is the first time it is being used for slip avoidance in dynamic manipulation tasks. The main difficulty relates to the lack of analytical models for tactile/slip dynamics. By the proposed constrained optimisation setting, we seek to benefit from the data-driven models for slip prediction and model-based approaches for online trajectory optimisation.

## 3  Methodology

Our proposed slip avoidance controllers are built in two stages: first, the offline slip classification phase where the LSTM-based networks (Section 3.1) are trained on a collected dataset (Section 4) and second, the online trajectory adaptation phase where the trained models are used in a constraint optimisation setting (Section 3.2) to minimise the expected value of the slip signal and deviation from a reference (desired) trajectory.

### 3.1  Slip detection and slip prediction models

Slip classification is explored by three main approaches in the literature: (i) incipient slip detection [10], (ii) full slip detection [13, 4], and (iii) slip prediction [5]. Incipient slip detection requires precise slip calibration and sensor mechanical compliance [7, 26]. Our focus is on using data-driven approaches for full slip detection and slip prediction. Tactile shear forces have proven to be a useful

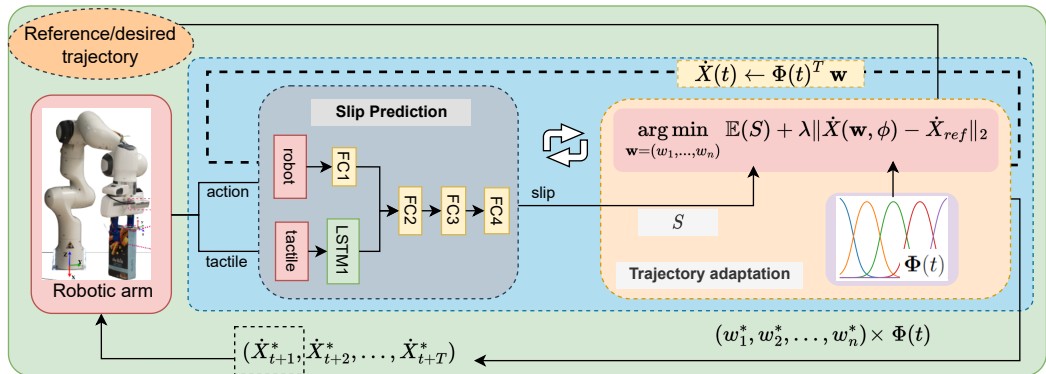

Figure 1: A schematic of the proposed *proactive* slip control (blue block) that is running at 30 Hz and includes a slip prediction module (grey block), and predictive controller for trajectory adaptation (orange block).

feature for slip classification [10]. This grounds the higher rate of changes for shear force when the taxels are losing contact with the objects, which is classified as an anomaly versus a stuck phase. [27, 28] show the benefits of using a history of tactile features and temporal processing by modules such as LSTM cells. As such we use the shear forces of the uSkin tactile sensor (introduced in Section 4) with two LSTM-based models for slip *detection* and *prediction* problems. In *slip detection*, a time window of length $C$ of shear tactile forces $\mathbf{x}_{t-C:t} \in \mathbb{R}^{C \times 48}$ are used as input features to classify the binary slip class $c_t = f(\mathbf{x}_{t-C:t})$ where $c_t \in \{c_{slip}, c_{non-slip}\}$ at time step $t$. For *slip prediction*, we use a multi-modal LSTM-based model (inspired by [11, 28]). This model uses time windows of tactile features $\mathbf{x}_{t-C:t} \in \mathbb{R}^{C \times 48}$ and robot future actions $\mathbf{a}_{t+1:t+T} \in \mathbb{R}^{T \times 6}$ to classify the slip state $c_{t+T}$ at time step $t + T$ (where $T$ is the prediction horizon). The architecture of the *slip prediction* model is shown in Fig. 1. The *slip detection* model has a similar architecture without the robot data input and the corresponding dense layer. Fusing the latent vectors of two input modalities seeks to exploit the future robot motion for slip prediction.

## 3.2 Closed-loop slip control by trajectory adaptation

The studies in closed-loop control design for slip avoidance are limited to gentle short-range lifting motions [4, 20, 29]. Our proposed approach extends slip control beyond simple slow lifting motions. First, we collected a data set of dynamic moving tasks by pre-defined task space trajectories, i.e. a linear motion with various velocity profiles and ranges of acceleration/deceleration values. We learned slips are initiated by the dynamic motions of the robot as opposed to being caused by object weight or a non-stable grasp for slow lifting tasks in literature [4, 20, 29]. Therefore, we propose two trajectory optimisation/adaptation methods incorporating tactile and robot proprioceptive data for slip control. We tested/compared PSC and RSC approaches in real-time scenarios on real robot experimentation.

**Reactive slip control by trajectory adaptation** Inspired by episodic policy search in [30], we represent the robot actions (*e.g.* task space velocity) by linear Gaussian parametric models. As such, robot's action at time $t$ can be represented as $\dot{X}_t := \sum_{j=1}^{N} w_j \phi_j(t)$, where $w_j \in \mathbb{R}$ are the weight parameters and $\phi_j$s are the basis functions with Gaussian forms $\phi_j(t) = \exp\left(-\frac{(t-\mu_j)^2}{2\sigma}\right)$ where $j = 1, 2, .., N$ is the number of basis functions.

A sequence of future robot actions with length $T$ in a matrix form will become $\dot{X}_{t+1:t+T} := \Phi^T \times W$ where $\Phi(t) = [\phi_1(t), \phi_2(t), ..., \phi_n(t)] \in \mathbb{R}^{N \times T}$ and $W = \{w_1, w_2, ..., w_n\} \in \mathbb{R}^N$ are the matrices of basis functions and weight values respectively. The mean values of the Gaussians are spread with equal distance in the horizon length $1 : T$ and the $\sigma$ parameters are chosen such that Gaussians have reasonable widths and overlap.

The main policy in grip force adaptation for slip control is to incrementally increase the normal force ($N(t)$) until the output of the slip detection model ($S$) is zero [28]. This is usually formalised as $N(t+1) = N(t) + S \times \delta N$, where $\delta N$ is the force increasing step. Likewise, we implement RSC by

adapting the robot movements trajectory by decreasing task space velocity by fine steps (we learned slip happens during the initial parts of movements with high acceleration) until the output of the slip detection model is zero. If no slip is detected, the robot will follow the desired reference trajectory. We empirically come up with a formulation in Eq. 1 after testing different ones. This includes an affine combination of slip cost and reference trajectory tracking cost. Although in RSC we do not need to optimize over a receding horizon, this constrained optimisation will be later extended in PSC (in Eq. 2) allowing easy comparison between *PSC* and *RSC*.

$$\underset{\mathbf{w}}{\arg\min} \quad \|\dot{X}_{t+1:t+T}(\mathbf{w}, \phi) - \dot{X}^{ref}_{t+1:t+T}\|_2 + a^{\frac{1}{S+\epsilon}}\|\dot{X}_{t+1:t+T}(\mathbf{w}, \phi)\|_2 \qquad (1)$$
$$subject\ to: \quad lb < \dot{X}_{t+1} - \dot{X}^{obs}_t < ub$$

Where $\dot{X}$ is the generated trajectory by optimisation, $\dot{X}^{ref}$ is the reference trajectory, $S \in \{0, 1\}$ is the output of the slip detection model introduced in 3.1, $a$ and $\epsilon \in \mathbb{R}^+$ are hyper-parameters, and $\dot{X}^{obs}_t$ is robot's observed velocity at time $t$. The first term in the objective function aims to minimise the Euclidean distance between the generated and the reference trajectory. The second term in Eq. 1 results in generated smaller velocities (where the object is moved with a large acceleration). As such, by choosing $a$ to be in $0 < a < 1$ and $\epsilon$ as a small positive value *i.e.* 0.01, for non-slip mode when $S = 0$, the effect of the second term vanishes ($a^{\infty}$ , $a \in (0, 1)$) and if $S = 1$, it constantly reduces robot's task space velocity until there is no slip. The imposed constraint keeps the robot action at $t$ bounded to avoid abrupt deceleration and ensures the admissibility condition of the robot's low-level controller is met. We selected the exponential term for the slip signal in that it resulted in more stable optimisation and better slip avoidance behaviour than other forms, e.g., linear cost of slip. The hyper-parameters are empirically chosen as $a = 0.5$ and $\epsilon = 0.01$.

**Proactive slip control by trajectory adaptation** Predictive controllers, e.g. [31], showed success when dealing with nonlinear systems with the risk of being stuck in a local minimum of the corresponding utility function. They can achieve a solution close to the global optimum by finite-horizon optimisation. We use the same approach by solving the constrained optimisation in a prediction horizon and applying the first component of the generated action sequence at each time step.

Parametric action representation has benefits in exploration complexity versus direct search in action space [30]. The benefit is more effective when the action is optimised in a future time horizon rather than a single time step. In this regard, in the trajectory adaptation problem, we use linear Gaussian parametric representation [32] for the robot action sequence. This leads to fewer search parameters in the optimisation problem relative to searching in the action space. In this scenario, by the *slip prediction* model introduced in section 3.1, we close the loop for PSC by solving the constraint optimisation in Eq. 2. The predicted slip value at time $t$ is a function of robot's future trajectory, $S(\dot{X}_{t+1:t+T}(\mathbf{w}, \phi))$.

$$\underset{\mathbf{w}}{\arg\min} \|\dot{X}_{t+1:t+T}(\mathbf{w}, \phi) - \dot{X}^{ref}_{t+1:t+T}\|_2 \qquad (2)$$
$$subject\ to: \qquad \mathbb{E}[S(\dot{X}_{t+1:t+T}(\mathbf{w}, \phi))] = 0$$
$$lb < \dot{X}_{t+1} - \dot{X}^{obs}_t < ub$$

Our PSC aims at minimising the expected slip over the prediction horizon while keeping close to the given reference trajectory as shown in Fig. 1. In contrast to RSC in Eq. 1, we consider the slip term as a non-linear constraint in Eq.2. This forms a more effective optimisation for slip minimisation since the optimisation library we use forms a Lagrangian function by optimising the Lagrange multipliers as opposed to the heuristic form of the multiplier in the second term of Eq.1.

## 4 Data set

### 4.1 Tactile sensor and manipulation task

Spatial resolution, reading frequency, and mechanical compliance are the main features of a tactile sensor reviewed in [33]. We use uSkin, a magnetic-based tactile sensor from XELA ROBOTICS

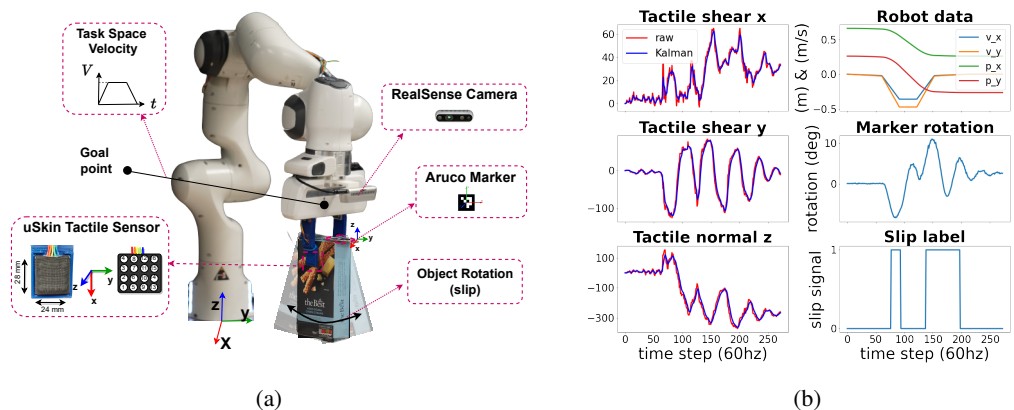

(a)                                                    (b)

Figure 2: (a) Experimental setup including Franka Robot, 4x4 uSkin tactile sensor on its fingers, box-shaped object, ArUco marker on the object, and RealSense camera. A sample of trapezoidal task space velocity trajectory is shown on the top left. (b) Data set features: tactile readings, robot trajectories, marker trajectory, and slip signal.

that works with 100 Hz frequency. It publishes 3-axis point-wise force measurements for 16 taxels arrayed in a 4x4 matrix (Fig. 2a). A pair of sensors are mounted on 3-D printed fingers for a jaw gripper. We use a box-shaped object ($4 \times 15 \times 25$ [cm] and 400 [gr]) for training and to encourage more slip-cases, we reduce the friction coefficient by covering the uSkin sensor with a *latex* cover and increase the object weight with large metal bolts fixed inside the box that do not move during manipulation. A wrist camera reads the pose of the ArUco marker attached on the top side of the object measuring the relative motion of the object w.r.t the hand. This is used to create labels for ground truth slip values. Maximum noise amplitude for marker readings is 0.38 [degrees] for rotation and 0.5 [mm] for the position in this range of distance.

The robot grasps the object from a top grasp pose with the jaw gripper, lifts it for 10 [cm] (in $+Z$ direction), and executes a linear motion in the horizontal ($XY$) plane. The linear motion encourages slip events which are a direct outcome of dynamic manipulation forces rather than object static weight. As such, for trajectories with small accelerations, the object keeps its upright orientation; While high accelerations cause the object's large rotation (as a result of the object's centre of mass moment of inertia around the grip axis) and slip. We use task space velocity controller with several reference trajectories with a trapezoidal profile and various maximum velocity ($V_{MAX}$) values. The velocity commands are converted to joint torques through the robot's low-level joint impedance controller. Fig. 2a shows a sample of the reference velocity profile (start and end point of the motion are $[0.4m, 0.25m, 0.3m]$ and $[0.1m, -0.25m, 0.3m]$ respectively in robot's base frame); As $V_{MAX}$ changes, the acceleration and deceleration values change accordingly such that the area under the curve which is equal to the overall translation remains constant.

## 4.2 Data collection and pre-processing

We collected a data set of 660 linear motion tasks containing (i) tactile readings from two fingers (at 100 [Hz]), (ii) robot state data (at 1,000 [Hz]) including joints position, joints velocity, task space pose, and task space velocity, (iii) Marker pose data (at 60 [Hz]), and (iv) Slip label values based on marker's rotation (see Fig. 2b). We use ROS *ApproximateTime* policy to synchronise the data. We applied Kalman Filter to the raw tactile readings to delete the measurement noise on the data (See Fig. 2b). Considering the linear robot motion, slip can happen either as pure rotational slip, or rotational slip followed by translational slip (which usually leads to object dropping and task failure). As such, a threshold on the rotation of the ArUco marker is used to create the slip labels. Based on marker rotations for slower ranges of motions with no slip, rotations $> 6°$ are labeled as slip-case. The data set along with preprocessing and models will be *publicly available* for future investigations of action-conditioned tactile or slip prediction [34, 35] or slip detection models in a dynamic manipulation task.

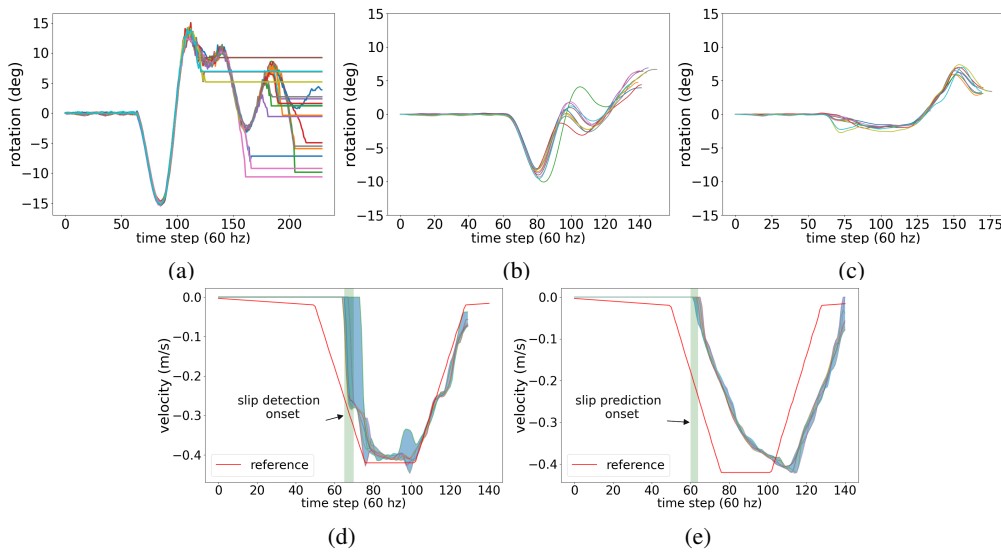

(a)  (b)  (c)

(d)  (e)

Figure 3: Object rotation around grasp axis during following a reference trajectory (a) without slip control and with (b) RSC, and (c) PSC, where $> 6°$ are slip events. Trajectories generated by RSC (d) and PSC (e): the blue areas show variations across 10 trails; the green bands show approximate time slip is detected (predicted) for the first time.

## 5  Experimental results and discussion

We tested the proposed trajectory adaptation methods to control slip in real-time in several test cases of manipulative trajectories with the Franka Emika robot connected to a PC with AMD® Ryzen 16-core processor and 64 GB RAM running ROS Noetic. We use the ONNX library to reduce the inference time of the slip classification PyTorch models from 1 [ms] to 0.2 [ms]. Scipy optimisation library in python 3 (i.e. *minimise()* function with *'trust-constr'*) is used to solve the constrained optimisation in Eq. 1 and 2. Each test trial is executed 10 times on the robot to achieve statistically meaningful results.

The metrics to compare the performance of slip controllers are (i) Resulted optimality value (ROV) from the optimisation library (*i.e.* gradient of the Lagrangian function, and indicative of the convergence of the optimisation), (ii) Maximum amount of object rotation (MOR) in the trial, (iii) Execution time (ET) in millisecond which shows the computation complexity, (iv) instances of rotations larger than $6°$, "RTS" (Rotation $> 6°$ Time Steps) score which is the number of time steps that the object had rotations larger than 6 degrees. Larger rotations and large number of cases of RTS indicate a higher probability of failure, and (v) distance to reference trajectory (DRT) which is the sum of Euclidean distance between the reference and adapted trajectories over the task execution time. In total, 300 testing trials are executed with the train and test objects. Slip classification accuracy and f-score on the test set (20% of the data) are 98% and 0.81 for the detection and 94% and 0.74 for the prediction models, respectively. The planning horizon $h$ is chosen by trading off between classification scores and controller success rate. While longer horizons worsen the classification performance, they give the proactive controller more reaction time and increase its success rate by having a larger safety margin. By testing horizons of lengths 5, 10, 15, and 20, we picked 10 as the best case. For the context frames length $C$, considering the sensory data frequency (60 hz) and task completion time (1.2-3 seconds), we chose $C = 10$ as a reasonable value including past 0.17 seconds of tactile data.

**Reactive versus proactive slip control with trajectory adaptation:** As the first test case, we compare RSC and PSC with the same reference trajectory and design parameters. Resulted object rotation for RSC and PSC are presented in Fig. 3b and Fig. 3c respectively; which can be compared to the case of executing the reference trajectory without a slip control in Fig. 3a (horizontal lines in Fig. 3a show the time instance the object is dropped off the robot's hand due to slip). RSC activates after slip is observed in reality (*i.e.* rotation $> 6°$ around time step 70 in Fig. 3b), but it keeps the

| Model | # Basis Functions | ROV | MOR | ET | RTS | DRT |
|---|---|---|---|---|---|---|
| Reactive | 2 | $0.80 \pm 0.72$ | $12.80 \pm 7.58$ | $9.79 \pm 1.33$ | $35 \pm 8$ | $1.31 \pm 0.15$ |
| | 3 | $0.71 \pm 0.56$ | $9.06 \pm 0.56$ | $10.03 \pm 1.57$ | $31 \pm 8$ | $1.20 \pm 0.15$ |
| | 4 | $0.67 \pm 0.45$ | $9.65 \pm 1.50$ | $10.49 \pm 1.62$ | $30 \pm 5$ | $1.16 \pm 0.22$ |
| | 5 | $0.63 \pm 0.31$ | $\mathbf{8.82} \pm 0.61$ | $9.96 \pm 1.49$ | $\mathbf{28} \pm 7$ | $1.04 \pm 0.23$ |
| | 6 | $0.47 \pm 0.23$ | $9.09 \pm 1.19$ | $10.15 \pm 1.70$ | $32 \pm 10$ | $1.05 \pm 0.19$ |
| | 7 | $0.37 \pm 0.22$ | $9.66 \pm 1.17$ | $\mathbf{9.70} \pm 2.66$ | $33 \pm 6$ | $1.06 \pm 0.17$ |
| | 8 | $\mathbf{0.28} \pm 0.19$ | $9.80 \pm 0.75$ | $9.99 \pm 1.50$ | $33 \pm 10$ | $\mathbf{0.99} \pm 0.21$ |
| Proactive | 2 | $0.75 \pm 0.60$ | $\mathbf{3.99} \pm 0.58$ | $\mathbf{11.73} \pm 5.35$ | $\mathbf{0} \pm 0$ | $1.67 \pm 0.05$ |
| | 3 | $0.72 \pm 0.67$ | $6.14 \pm 1.52$ | $22.57 \pm 1.79$ | $4 \pm 2$ | $1.56 \pm 0.24$ |
| | 4 | $0.51 \pm 0.41$ | $7.16 \pm 1.07$ | $25.73 \pm 2.14$ | $5 \pm 2$ | $1.49 \pm 0.27$ |
| | 5 | $0.40 \pm 0.34$ | $9.06 \pm 0.55$ | $29.61 \pm 3.26$ | $8 \pm 4$ | $1.26 \pm 0.20$ |
| | 6 | $0.33 \pm 0.22$ | $9.75 \pm 0.65$ | $32.09 \pm 5.05$ | $13 \pm 5$ | $1.23 \pm 0.18$ |
| | 7 | $0.30 \pm 0.22$ | $9.41 \pm 0.55$ | $35.27 \pm 3.65$ | $11 \pm 3$ | $\mathbf{1.17} \pm 0.36$ |
| | 8 | $\mathbf{0.25} \pm 0.20$ | $9.14 \pm 0.37$ | $37.55 \pm 3.84$ | $10 \pm 6$ | $1.21 \pm 0.21$ |

Table 1: Trajectory optimisation performance by different number of basis functions. For all of the metrics, smaller values show better performance. (all metrics are defined in Section 5)

rotation bounded in the rest of the trial. PSC (shown in Fig. 3c) outperforms RSC in the acceleration phase of the motion as the slip control activates before slip actually initiates. Both controllers show 100% success in avoiding task failure while in the mode without slip controller (Fig. 3a) 19 trials out of 20 led to object dropping and failure.

The velocity profiles generated by RSC and PSC are shown in Figs. 3d and 3e. Although these figures show RSC keeps closer than PSC to the reference trajectory, it yields much more rotation of the object and non-smooth (with a higher jerk that is undesirable) movements. PSC yields trajectories farther from the reference trajectory but has smoother acceleration and deceleration trends which can lead to smaller slip instances. This is mostly because the weight of the slip avoidance relative to the Euclidean distance term in Eq. 2 is set by the Lagrange multipliers of the optimiser in PSC. This makes the trajectories get more distant from the reference trajectory to satisfy the slip constraint.

**The effect of the number of basis functions on trajectory adaptation:** We tested *RSC* and *PSC* (*Reactive* and *Proactive*, respectively, in Table 1) with number of basis functions ranging from 2 to 8. While PSC sees a prediction of slip it has enough time to deviate enough from the reference to avoid slip more effectively. This leads to having smaller RTS and larger DRT for PSC. On the other hand, RSC has a shorter reaction time relative to PSC, which leads to larger RTS and smaller DRT. In terms of improving RTS (slip prevention performance), maximum improvement in RSC by changing the number of basis functions is $\frac{35-28}{35} \times 100 = 20\%$. While switching from RSC to PSC with 5 basis functions for both, improves slip prevention by $\frac{28-8}{28} \times 100 = 71\%$. This improvement can be higher for other numbers of basis functions.

Depending on how many basis functions are used, the overlapping area of the first Gaussian with the next ones can change (since we take only the first element of the generated action sequence in the horizon, the change in the beginning part of the horizon is most important). For RSC, this change has a positive effect on having smaller RTS up to 5 basis functions. For reference tracking (DRT), increasing the number of basis functions improves tracking performance. For PSC, 2 basis functions resulted in zero RTS with the largest distance to the reference. Since the controller has fewer degrees of freedom in this case, the cost it needs to pay for preventing slip is to get much farther from the reference (the generated trajectories can be seen in Figure 2 (a) in the Appendix). This is a very conservative proactive controller and based on how much weight we want to put on slip prevention w.r.t reference tracking, we could prefer this controller. In fact, the number of basis functions is a control parameter that we can use to specify how much conservative we want to be about slip avoidance. The safety margin is achieved by reducing the number of basis functions, but the cost is getting farther from the reference trajectory. Changing the basis functions from 2 to 5 for PSC, improves tracking performance by 6%, 4%, 15% respectively, but going further to 6 basis functions has a large negative effect on RTS ($\frac{13-8}{8} \times 100 = 62\%$).

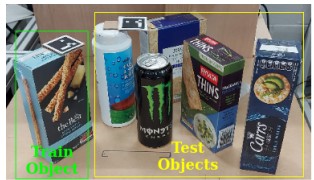

Figure 4: Train and test objects.

| Object | ROV | MOR | RTS | DRT |
|---|---|---|---|---|
| theBest | $0.51 \pm 0.41$ | $7.16 \pm 1.07$ | $4 \pm 2$ | $1.53 \pm 0.2$ |
| THINS | $0.54 \pm 0.43$ | $7.88 \pm 1.38$ | $8 \pm 3$ | $1.71 \pm 0.1$ |
| Carrs | $0.52 \pm 0.39$ | $8.02 \pm 2.12$ | $7 \pm 3$ | $1.61 \pm 0.3$ |
| RICE | $0.61 \pm 0.44$ | $10.58 \pm 3.56$ | $10 \pm 2$ | $1.51 \pm 0.2$ |
| Monster | $0.66 \pm 0.46$ | $36.20 \pm 13.93$ | $28 \pm 5$ | $1.24 \pm 0.3$ |
| Slug killer | $0.52 \pm 0.41$ | $8.23 \pm 1.86$ | $6 \pm 2$ | $1.62 \pm 0.3$ |

Table 2: Tests on novel objects.

As such we choose 5 basis functions to be the best value for PSC based on this quantitative analysis. While ROV improves by an increased number of basis functions for both controllers, MOR achieved by RSC does not show a clear improvement after 5 basis functions. In contrast, it leads to a larger MOR for PSC. The computation time increases proportionally in PSC, while it remains roughly constant in RSC. The reason for the larger computation time for PSC is that the Jacobians of the slip constraint should be numerically estimated by the optimisation library which needs 30-50 calls of the slip detection/prediction networks in each optimisation iteration.

**Generalisation tests with novel objects:** To evaluate generalisation on novel tactile features we test PSC with 5 basis functions on the novel objects shown in Fig. 4 (from left to right the objects are called theBest, Slug killer, Monster, RICE, THINS, and Carrs respectively). Three of the objects have a box shape similar to the training object, and two are cylindrical for more difficult generalisation cases with novel contact geometry. Table 2 shows the resulted metrics for the train (first row) and the test objects. The first two test objects have similar geometry and weight to the train object but with different friction behaviour where the PSC performance is close to the ones with the train object in terms of both RTS and MOR. Although the RICE box has similar geometry, the PSC performance is relatively poor. This is partially related to the lighter weight and different friction behaviour of the box cover. We observed our PSC can successfully avoid slip in manipulating objects unseen in the training.

The Monster can with dynamic weight (liquid inside), deformable surface, and different geometry (cylindrical) and friction is the most challenging test object for the controller and resulted in the worst performance metrics values. The PSC can effectively control the Slug Killer can slip which has a plastic and rigid material very different from the train object. The results in Table 2 demonstrate PSC can generalise to a novel contact geometry for a rigid object. In fact, the proposed constrained optimisation setting is dependent on how reliable the slip detection/prediction model performs. As such, as far as the slip classification model can generalise to novel objects, the proposed slip avoidance control is guaranteed to effectively avoid object slip and task failure.

## 6   Conclusion

We presented a novel approach to slip control by online trajectory adaptation. This is very important as the increased gripping force control policy is not an effective solution in many settings. We presented two novel approaches for slip control by robot trajectory adaptation (i) Reactive Slip Control and (ii) Proactive Slip Control. We showed the success of the slip controllers in real-time tests on a dynamic real-world manipulation task. We show Proactive Slip Control outperforms the reactive one and it can generalize to objects unseen in training. Our future works include extending our dataset to include different tactile readings, e.g. camera-based tactile readings, and more real-world manipulation experiments, i.e. different objects and manipulative movements.

**Limitations:** Our approach is heavily relying on the slip data set to predict the slip. We have only one class of trajectories (i.e. a path with different velocity profiles) in our dataset which limits the prediction performance. We will extend the dataset in our future works and collect data with different classes of manipulative movement trajectories. The training dataset only includes one object with varying weights. Nonetheless, the slip controller could successfully generalise to unseen objects in our test cases. We will improve our dataset by adding more objects (from multiple classes) to our dataset in our future works. The controller is using one predicted slip signal to adjust its next action. Using all the time steps in a prediction horizon may improve the performance. In our future works, we will use a classifier for different predicted tactile readings provided by an action-conditioned tactile predictive model to improve the performance of slip control.

**Acknowledgments**

This work was partially supported by Centre for Doctoral Training, United Kingdom (CDT) in Agri-Food Robotics (AgriFoRwArdS) Grant reference: EP/S023917/1; Lincoln Agri-Robotics (LAR) funded by Research England; and by ARTEMIS project funded by Cancer Research UK C24524/A300038.

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
