# OpenReview forum: "Proactive slip control by learned slip model and trajectory adaptation"
_robot-learning.org/CoRL/2022/Conference — CoRL 2022 Poster_

### Official Review · Reviewer_b9H8 · 2022-07-19

**Originality:** Fair
**Technical Quality:** Good
**Clarity Of Presentation:** Fair
**Impact:** 3

**Recommendation:**

Weak Accept: I recommend accepting the paper, but will not argue for my recommendation if the majority of other reviewers have a different opinion.

**Summary:**

The paper develops a novel algorithm for contact slip avoidance in robotic manipulation of objects. The algorithms consists of an offline procedure of learning a slid detection and prediction model and online trajectory adaptation to prevent slip. Several experiments are conducted.



**Issues:**

The issues to be addressed consist of:
-  Lack of good flow and structure, I believe a section before Section 3. summarizing the flow of the proposed algorithm, breaking down the offline learning and online optimization would be very beneficial.
- Lack of comparative experiments with respect to related works.

Further, why is the robot's action parameterized by a Gaussian parametric model? Why not just consider the robot action as the optimization variable? Similarly, the notation $\dot{X}$ (without subscripts) seems to be undefined.

The paper should be revised with respect to some grammatical mistakes.

**Quality Of The Limitations Section:**

Limitations are addressed clearly

**Reviewer Expertise:**

3: The reviewer is fairly confident that the evaluation is correct

**Robotics Focus:**

Sufficient demonstration on hardware

**Strengths And Weaknesses:**

Strengths:
- The paper deals with an interesting and relevant problem.
- The proposed method seems to be novel with respect to the related literature.
- There are numerous experimental results to test the method.

Weaknesses:
- In some parts, the paper lacks clarity and good structure. The most important is that the overall algorithm is unclear from a first readthrough.  I understand that two models are trained using offline trajectories, one for slid detection and one for slip prediction. Then these models are used in the online optimization process. However, this is not explicitly mentioned in the paper.
- Since there are not dynamics taken into account and no theoretical guarantees, the proposed algorithm should be tested with other state-of-the-art algorithms on slip prevention.



**Summary Of Recommendation:**

The paper seems to propose a novel algorithm, but it needs further revision on the structure and presentation as well as the experimental results.

---

> ### Author Response · Authors · 2022-08-22
> **Author Response to Reviewer b9H8**
>
> Dear Reviewer b9H8, we thank you for your detailed and thorough review.
>
>
> ` In some parts, the paper lacks clarity and good structure. The most important is that the overall algorithm is unclear from a first readthrough `
>
> We listed the revisions we did on the manuscript in **General Response 3**. We believe these revisions substantially improved the clarity and quality of the paper.
>
> ` Since there are not dynamics taken into account and no theoretical guarantees, the proposed algorithm should be tested with other state-of-the-art algorithms on slip prevention. (Lack of comparative experiments with respect to related works.) `
>
> We discuss the strategy we had to perform stability analysis for the proposed slip controllers in **General Response 1**.
>
> We fully agree comparing to other slip prevention methods in the literature could further clarify the effectiveness/position of our trajectory adaptation method w.r.t the grip adaptation approaches. We originally aimed to do so. However, we realised Franka Emika Panda robot/gripper which is widely used in academia and industry nowadays does not allow grip force control during manipulative movements (the gripper has blocking TCP/IP communication, and it cannot be controlled in real time). Hence, the object is grasped a the beginning of the task with constant (uncontrolled) grip force before manipulative movement starts.
>
> In fact, we started the project thinking of grasp adaptation as the slip prevention method; however, after learning our hardware limitation, and knowing the facts that increasing grip force fails in scenarios where the object is delicate/soft or the gripper already reached its maximum force, motivated us to come up with an approach for slip prevention which could overcome these limitations. As such, although both proposed trajectory adaptation methods (RSC and PSC) are novel slip prevention approaches since RSC is motivated by slip prevention methods in the literature (the only difference is we replace grip force by task space velocity as control input), we considered it as our baseline model for comparisons.
>
> ` Lack of good flow and structure, I believe a section before Section 3. summarizing the flow of the proposed algorithm, breaking down the offline learning and online optimization would be very beneficial. `
>
> We address this comment in the listed revisions in **General Response 3**.
>
> ` why is the robot’s action parameterised by a Gaussian parametric model? Why not just consider the robot action as the optimisation variable? `
>
> Solving an optimisation problem with multiple input/output in real-time is computationally expensive. As a grounded solution to this problem, proposed by Robot Learning community, the continuous space is discretized with basis functions to reduce the optimisation parameters [1, 2]. One of the common discritization methods is by using Gaussian basis functions as we did in the paper. As an example in our problem, when prediction horizon length is h = 20, with 6 dof Cartesian velocity, in the case of action space we will have $20 \times 6=120$ optimisation parameters. While for the parametric case, by defining 4 basis functions for each dof, the number of optimisation parameters will be $4 \times 6=24$ weight values. This will lead to easier optimisation for parametric action representation.
>
> ` Similarly, the notation X seems to be undefined. `
>
> In the revised version we introduce $\dot{X}$ in Section 3.2 "Close-loop slip control by trajectory adaptation."
>
> ` The paper should be revised with respect to some grammatical mistakes. `
>
> The paper is carefully read and corrected multiple times for grammatical mistakes.
>
>
>
> &nbsp;
>
> *[1]. Schaal, Stefan. "Dynamic movement primitives-a framework for motor control in humans and humanoid robotics." In Adaptive motion of animals and machines, pp. 261-280. Springer, Tokyo, 2006.*
>
> *[2]. Deisenroth, Marc Peter, Gerhard Neumann, and Jan Peters. "A survey on policy search for robotics." Foundations and Trends® in Robotics 2, no. 1–2 (2013): 1-142.*

---

> > ### Comment · Reviewer_b9H8 · 2022-08-24
> > **Additional comments**
> >
> > I thank the authors for the response.
> > Is there a revised version of the paper uploaded? If not, please upload one so that the revision can be checked by the reviewers.
> >
> > Further, slip prevention via trajectory adaptation and considering the task-space velocity as control input is definitely interesting; however, not being able to compare with other approaches where the grip force is actually controlled is a big limitation, since grip force is the main component for such a property.  Additionally, reducing task-space velocity for slip prevention can result in unnecessarily conservative motions - which could be avoided by a potentially strong grip force.

---

> > > ### Author Response · Authors · 2022-08-27
> > > **We implemented a grip force control and did a comparison study**
> > >
> > > **Comment:**
> > >
> > > Dear Reviewer b9H8, we thank you for your valuable comments.
> > >
> > > `Is there a revised version of the paper uploaded?`
> > >
> > > Please find the revised version of the paper with this [link](https://drive.google.com/file/d/1fLZIeL6G_6L3S-uFcyEakUYDBLRwiTs1/view?usp=sharing).
> > >
> > > `however, not being able to compare with other approaches where the grip force is actually controlled is a big limitation, since grip force is the main component for such a property. `
> > >
> > > According to your comment, we implemented a grip force control (GFC) [1, 2] using an SMC parallel Jaw Gripper. We compared the results obtained by GFC and our proposed trajectory adaptation methods for slip prevention. The grip force applied by SMC parallel gripper increases every time the slip detection model predicts a positive value by tightening the grip. The results are shown in the table included in the attached file (for both train and novel objects). MOR (Maximum Object Rotation) and RTS (Rotation$>6^{\circ}$ Time Steps) show object's maximum rotation and number of time steps in task execution time where object has rotations larger than $6^{\circ}$ respectively. According to these two metrics, Proactive Slip Control (PSC) outperforms the GFC with respect to slip prevention and keeping objects' rotation bounded.
> > >
> > >
> > > We also attached the video of the experiments. Furthermore, we present the performance of the controllers for the train object in the figure in the attached file. Although the grip controller fails in the acceleration phase (due to objects high rotational moment) it can fully stabilize the object in the deceleration part of the motion. Overall, PSC shows the best performance w.r.t keeping object rotation bounded (less than 6). The results are added to the revised version of the paper (which is now more than the page limit (this was allowed by the conference only in the rebuttal phase), but we will reduce the pages to 8 pages in case of being accepted). Comparing a proactive grip force controller with a proactive trajectory adaption method is also an interesting case which we will explore in the future work.
> > >
> > > `reducing task-space velocity for slip prevention can result in unnecessarily conservative motions - which could be avoided by a potentially strong grip force`
> > >
> > > Our objective functions in equation (1) and (2) penalize the distance of the generated trajectory from the reference trajectory; Since the reference trajectory is a minimum-time trajectory the optimization avoids unnecessary slow motions as much as possible.
> > >
> > > &nbsp;
> > >
> > > [1]. Veiga, Filipe, Benoni Edin, and Jan Peters. "Grip stabilization through independent finger tactile feedback control." Sensors 20, no. 6 (2020): 1748.
> > >
> > > [2]. Khamis, Heba, Benjamin Xia, and Stephen J. Redmond. "Real-time Friction Estimation for Grip Force Control." In 2021 IEEE International Conference on Robotics and Automation (ICRA), pp. 1608-1614. IEEE, 2021.
> > >
> > > **Zip File:**
> > >
> > > /attachment/097f28f9646d6c0bd3938b51076c57bd076e05ca.zip

---

### Official Review · Reviewer_xX6y · 2022-07-24

**Originality:** Good
**Technical Quality:** Good
**Clarity Of Presentation:** Fair
**Impact:** 3

**Recommendation:**

Weak Accept: I recommend accepting the paper, but will not argue for my recommendation if the majority of other reviewers have a different opinion.

**Summary:**

This paper applied machine learning to the problem of slip detection and avoidance in manipulation tasks. They implement two solutions: one a purely reactive method which uses an LSTM to detect slip events triggering the controller to slow it's velocity. The second implementation is also an LSTM but takes in both past data from the tactile sensor and future planned actions from the controller to predict slip. These future predictions are then used to re-optimize the planned trajectory to pre-emptively avoid slip.

**Issues:**

- The biggest issue with this paper is that the experimental setup and reference trajectory is not explained. It was not clear at all why rotation was used to estimate slip until watching the video. The authors should clearly explain the goal trajectory used in the experiments and how exactly does the object slip out of the robot's grasp. From looking at Figure 1, I assumed the robot just lifted the box in the air which made the entire paper very confusing.
- The text in Figure 1 is too small and blurry. Also adding labels for all the signals would make it more clear. For example writing "actions" and "tactile data" next to the arrows coming from the robot.
- I find the order of the paper confusing, perhaps the datasets section should go after Methodology.
- Figure 3 should include the object names either in the figure or caption so readers can know which object is theBest vs THINS for example.
- Some important implementation details are missing such as planning horizon and tactile data window length.

**Quality Of The Limitations Section:**

Limitations are addressed clearly

**Reviewer Expertise:**

3: The reviewer is fairly confident that the evaluation is correct

**Robotics Focus:**

Sufficient demonstration on hardware

**Strengths And Weaknesses:**

Strengths:
- Good application of learning. Slip is very difficult to model and learning methods are becoming more common for this purpose.
- Well justified choices in network and problem formulation

Weaknesses:
- The paper is unclear on some important information such as experimental setup.
- I generally find the paper is written in a confusing way with implementation details in the beginning and theory later resulting in a lot of back and forth reading.

**Summary Of Recommendation:**

I think this paper is a solid idea with good implementation. The weakest parts are in the writing of the paper itself which I expect the authors can improve. I have listed my issues with the paper in it's current form below.

---

> ### Author Response · Authors · 2022-08-22
> **Author Response to Reviewer xX6y**
>
> Dear Reviewer xX6y, we thank you for your detailed and thorough review.
>
> ` The biggest issue with this paper is that the experimental setup and reference trajectory is not explained. It was not clear at all why rotation was used to estimate slip until watching the video. `
>
> We listed the revisions we did to improve the clarity of the experimental setup and reference trajectory in **General Response 3** (updated figures are attached to the comment *Structure of The Responses* on top of the page). We also added a clear explanation of the linear reference trajectory in Section 4 "Data set" and specified the $[X, Y, Z]$ of start and goal point in robot's base frame (shown in Figure 1 (a) on robot's base) as $[0.4m, 0.25m, 0.3m]$ and $[0.1m, -0.25m, 0.3m]$.
>
> ` The text in Figure 1 is too small and blurry. `
>
> Figure 1 is updated as described in **General Response 3** and is attached to the comment *Structure of The Responses* on top of the page.
>
> ` Also adding labels for all the signals would make it more clear. For example writing "actions" and "tactile data" next to the arrows coming from the robot. `
>
> The updated version of Figure 1 (c) is attached to the comment *Structure of The Responses* on top of the page.
>
> ` I find the order of the paper confusing, perhaps the dataset section should go after Methodology. `
>
> We addressed this comment in **General Response 3**, and this substantially improved the flow and clarity of the paper.
>
> ` Figure 3 should include the object names either in the figure or caption so readers can know which object is theBest vs THINS for example. `
>
> The caption of Figure 3 is updated to include objects' names from left to right in order.
>
> ` Some important implementation details are missing such as planning horizon and tactile data window length. `
>
> We added details about the planning horizon and context frame lengths chosen in "Experimental results and discussion". The planning horizon is chosen by trading off between classification scores and controller success rate. Longer horizons decrease the classification performance because the slip prediction models should classify slip states far in the future; On the other hand, long horizons would give the proactive controller more reaction time and increases its success rate by having a larger safety margin. By testing horizons of lengths 5, 10, 15, and 20, we picked 10 as the best case. For the context frames length $C$, considering (i) the sensory data frequency (60 hz), and (ii) task completion time (1.2-3 seconds), we chose $C=10$ as a reasonable value including past 0.17 seconds tactile data. The page limit does not allow to include this discussion in the paper but we will add it in the Appendix with corresponding quantitative and qualitative discussion for choosing the horizon and context lengths.

---

### Official Review · Reviewer_PezY · 2022-07-27

**Originality:** Very Good
**Technical Quality:** Good
**Clarity Of Presentation:** Very Good
**Impact:** 3

**Recommendation:**

Weak Accept: I recommend accepting the paper, but will not argue for my recommendation if the majority of other reviewers have a different opinion.

**Summary:**

The goal of this work is to develop a slip control mechanism for robot manipulation tasks where objects being manipulated may move relative to the gripper due to insufficient shear force. The key idea is to learn a slip prediction model that takes tactile sensor data and planned future actions as input and outputs an estimation of whether the object is going to slip. The learned slip prediction model is then used as a constraint in a trajectory optimization formulation to adjust the planned robot motion in order to proactively prevent slipping. The proposed model is trained with a single object with varying weights and robot motions and evaluated on novel objects. It is also compared to a baseline method of reactive slip control, where the robot motion is adapted after detecting the slip, and the robot motion without considering the slip. The proposed method achieved a comparable success rate compared to reactive slip control, while being able to reduce the object orientation.

**Issues:**

- What's the range of weights seen during data collection? It is mentioned that metal pieces are added to the box but I didn't find how many of them.
- How would the reactive slip control perform is a buffer is used, e.g. if slip is defined as exceed 6 degrees in orientation, the detection model can be trained to trigger when the orientation exceeds 4 degrees. The paper mentioned that RSC can already track the reference motion better, if setting the threshold more conservatively also improves orientation control it may perform at least as good as the proactive one?
- What are the statistics for the two terms in the customized optimality separately? It is mentioned that RSC tracks the motion better, which corresponds to the first term. Does it mean that slip happens more often in RSC?

**Quality Of The Limitations Section:**

Limitations are addressed clearly

**Reviewer Expertise:**

4: The reviewer is confident but not absolutely certain that the evaluation is correct

**Robotics Focus:**

Sufficient demonstration on hardware

**Strengths And Weaknesses:**

Strengths
- The idea of learning a predictive model for preventing slipping during manipulation is interesting and the proposed algorithm seems concrete.
- The collected dataset for slipping prediction/detection could be of interest to researchers in relevant fields.
- The result is demonstrated on real, physical robots with non-trivial manipulation tasks.

Weaknesses
- The advantage over reactive slip control doesn’t seem a lot
- The performance of the proposed method (PSC) seems to get worse as the number of basis functions increases. This raises a concern for the ability of the algorithm to be applied to more complex tasks where a higher-order motion is needed.
- The designed customized optimality is not very straightforward. I think it’s more clear and simpler if they separately report the slipping events and the tracking performance, instead of merging them with some weights alpha.


**Summary Of Recommendation:**

As mentioned in the weaknesses above, my main concerns for the work is that it didn't show much advantage over the reactive slip control, which could also use some additional tweaking, and it's not clear how well it can be extended to more complex tasks given the experiment results. That being said, I think the idea of learning to proactively prevent slipping during manipulation is interesting, the dataset generated and the designed task in this work may inspire future research in this area. Thus I'm currently leaning towards accept.

---

> ### Author Response · Authors · 2022-08-22
> **Author Response to Reviewer PezY**
>
> Dear Reviewer PezZ, we thank you for your detailed and thorough review.
>
> ` The advantage over reactive slip control doesn’t seem a lot `
>
> The state-of-the-art in the literature are reactive grip force controllers for slip prevention [1, 2]. The main contribution of our paper is proposing two novel slip prevention methods (Reactive Slip Control (RSC) and Proactive Slip Control (PSC)) based on learned slip models and trajectory adaptation. Although RSC is novel in this context, since it has its design motivated by the grip force controllers, we considered it as the baseline model for trajectory adaptation. As such, although we demonstrate the advantage of PSC over RSC, this is not claimed to be the main contribution of the paper. In **General Response 2** we try to clarify the performance of each of the slip controllers and the advantage of PSC over RSC based on different numbers of basis functions by replacing the customised metrics with its sub-components (reference tracking and object rotation) metrics.
>
>
> ` The performance of the proposed method (PSC) seems to get worse as the number of basis functions increases. This raises a concern for the ability of the algorithm to be applied to more complex tasks where a higher-order motion is needed. `
>
> We address this concern with quantitative analysis in **General Response 2**. Based on the presented discussion, the number of basis functions can be used as a safety parameter to tune how conservative the controller can be about slip prevention and reference tracking. Also showing the performance does not drop with more basis functions. In the extension work of this paper which is described in **General Response 1**, we relaxed the 1 dof motion constraint and execute the optimisation for all 6 dof in real-time. We tested the model with different classes of reference trajectories (polynomials of different orders and trapezoidal) with novel start and end points. The results show with the new dataset, the proposed trajectory adaption methods can deal with more complex tasks and still do effective slip prevention.
>
> ` The designed customised optimality is not very straightforward. I think it’s more clear and simpler if they separately report the slipping events and the tracking performance, instead of merging them with some weights alpha. `
>
> As described in **General Response 2**, we replaced the customised optimality with its building components and this substantially clarified the analysis of the performance of each method.
>
> ` What's the range of weights seen during data collection? It is mentioned that metal pieces are added to the box but I didn't find how many of them. `
>
> The range of object weight can vary between 200-600 grams. The metal pieces are large metal bolts that are fixated inside the box and do not move inside the object during the experiments.
>
> ` How would the reactive slip control perform if the detection model was trained to trigger when the orientation exceeds 4 degrees. RSC can already track the reference motion better, if setting the threshold more conservatively also improves orientation control it may perform at least as good as the proactive one? `
>
> While PSC is trying to give response to an event far enough in the future, RSC does not have this amount of reaction time.  Based on what we learned, we expect by having $4^{\circ}$ threshold for RSC, the controller activates many more times causing hysteresis effect in our controller. Moreover, the larger number of controller activation, the larger value of integral of distance to reference trajectory. These two effects are not desired. Nonetheless, it is interesting and we will study the suggested scenario for RSC with $4^{\circ}$ threshold in our journal paper for choosing the optimal value for this threshold.
>
> ` What are the statistics for the two terms in the customised optimality separately? `
>
> The statistics can be found in the updated version of Table1 attached to the comment *Structure of The Responses* on top of the page.
>
> ` It is mentioned that RSC tracks the motion better. Does it mean that slip happens more often in RSC? `
>
> The number of slip instances in the RSC is higher than PSC (see RTS column in the updated version of Table 1). This is caused by (i) RSC activates after a couple of time steps of slip, which is not the case for PSC, and (ii) RSC has a shorter reaction time to slip events relative to PSC during the task execution.
>
> &nbsp;
>
> [1]. Veiga, Filipe, Benoni Edin, and Jan Peters. "Grip stabilization through independent finger tactile feedback control." Sensors 20, no. 6 (2020): 1748.
>
> [2]. Khamis, Heba, Benjamin Xia, and Stephen J. Redmond. "Real-time Friction Estimation for Grip Force Control." In 2021 IEEE International Conference on Robotics and Automation (ICRA), pp. 1608-1614. IEEE, 2021.

---

### Official Review · Reviewer_FJ4L · 2022-07-29

**Originality:** Very Good
**Technical Quality:** Very Good
**Clarity Of Presentation:** Very Good
**Impact:** 4

**Recommendation:**

Weak Accept: I recommend accepting the paper, but will not argue for my recommendation if the majority of other reviewers have a different opinion.

**Summary:**

This paper proposes reducing slip during dynamic motions by adapting the trajectory, specifically decreasing the velocity. This is in contrast to previous papers which normally focus on moderating gripping force. After gathering a dataset of slip experienced while completing linear motions with varying acceleration and deceleration profiles, the paper proposes two LSTM-based models: slip detection and slip prediction. These models are used in the two proposed methods: Reactive Slip Control (RSC) and Proactive Slip Control (PSC). RSC reactively decreases the task space velocity until slip detection gives zero while PSC formulates a trajectory optimization problem where the objective is to minimize expected slip while staying close to reference trajectory. Unsurprisingly PSC outperforms RSC, since it can plan to prevent slip before it occurs. Since trajectories are modeled by linear gaussian parametric models the paper compares how the number of basis functions impacts performance.

**Issues:**

It could strengthen the paper to discuss how this method could be applied for translational slip and how the system could handle both rotational and translational slip. If possible, it would be great to consider a larger set of training objects/trajectories, although it is not clear that this is feasible in the rebuttal period.

A few more minor notes, questions, comments:
- In Fig. 2 it would be easier to compare the rotations if all the y-axes were the same. Additionally, is it correct to assume that the red line in (d) and (e) corresponds to the reference trajectory?
- How is the time window C chosen for RSC and PSC? How would varying this window impact the results? Likewise, how was the prediction horizon h chosen? (For the experiments it seems h=10). How would varying this impact results?
- The explanation of RSC mentions that formulation was found empirically. Is this referring to the form of the cost function or the constants? What other formulations were considered and why was the presented form selected?
- Is there any intuition on why PSC performs better with a small (2) set of basis functions? In particular, are there thoughts on why RSC needs more (5) basis functions?
- Sec 3 mentions that uSkin can be challenging to calibrate. Given that this was mentioned, did this impact the results in any way and are there tricks that can be done to mitigate this issue?
- With respect to the training set object, it would be helpful to mention, perhaps in the appendix, what weights were used. Is it also correct to assume that the metal pieces were fixed in the box such that they could not move?
- It might be more beneficial to place the data set section after the methodology section, since doing so would provide more context for the data set section.
- On the first page, line 34 there is a missing close paren.

**Quality Of The Limitations Section:**

Limitations are addressed clearly

**Reviewer Expertise:**

4: The reviewer is confident but not absolutely certain that the evaluation is correct

**Robotics Focus:**

Sufficient demonstration on hardware

**Strengths And Weaknesses:**

This paper takes an interesting approach to reducing slip by modifying the trajectory rather than the gripping force. The paper argues that in several scenarios, such as manipulating delicate objects, it may be undesirable to simply grip harder. The paper offers both a reactive and proactive formulation to address slip. While the results show that a proactive approach leads to less overall slip, it would be interesting to discuss if there is ever a case where RSC would be preferable to PSC. There is a very interesting discussion in the appendix on when grip force and trajectory adaption might be necessary in combination.

Both RSC and PSC are well-explained. The video also does an excellent job of communicating the main points of the paper and showing off the results.

One slight weakness of the paper is that the training set is composed of only one object (with varying weights) and the only trajectories are linear motions (with varying acceleration and deceleration profiles). As shown in the experimental section, this limits the generalizability. In testing on novel objects, the maximum rotation for all objects is above the desired 6 degree threshold. A larger training set could tease out where the generalization issue stems from.

In considering slip, the paper ignores translational slip, only focusing on rotation slip (defined as a slip of more than 6 degrees). Is translation slip considered less of an issue? Would translation slip be a larger concern for a different class of motions? The presentation of results is more clear when focusing on just rotational slip. However, given the emphasis on dynamic motions, it is a limitation to ignore translational slip.

**Summary Of Recommendation:**

I am weakly recommended that the paper be accepted. The paper presents an interesting alternative to addressing slip and distinguishes itself by considering dynamic trajectories. The two proposed methods (RSC and PSC) are mathematically sound and there are interesting and relevant robotics experiments comparing them and analyzing their parameters. While the proposed methods could be more general, by leveraging a more diverse training set and considering translational slip, I believe the contributions are still compelling and well-demonstrated through the experimental section.

---

> ### Author Response · Authors · 2022-08-22
> **Author Response to Reviewer FJ4L, part1**
>
> Dear Reviewer FJ4L, we thank you for your detailed and thorough review.
>
> ` One slight weakness of the paper is that the training set is composed of only one object (with varying weights) and the only trajectories are linear motions (with varying acceleration and deceleration profiles). `
>
> We aimed to demonstrate a proof of concept of using trajectory adaptation instead of grip force control for slip avoidance. As such, the main variation we considered to have in the dataset was different acceleration/deceleration values to be able to analyse the performance of the proposed novel slip controllers in a semi-structured task. A detailed description of the extension of this paper is described in **General Response 1** where we explain how we addressed task/motion variation limitations. Including the dataset and experiments described in **General Response 1** in the paper is not within the context of the contributions of the paper considering the page limitation.
>
> ` In considering slip, the paper ignores translational slip, only focusing on rotation slip `
>
> The first task we tried in the project was an upward lifting motion with various acceleration values to encourage translational slip. The only slip instances in that task belonged to slower ranges of motions as a result of object's static weight rather than dynamic manipulation forces. This also gave us very few slip cases and a highly imbalanced classification problem. After learning that, we considered a task where slip is encouraged by the rotational moment of object's centre of mass around grip axis. In the new dataset introduced in **General Response 1**, there are translational slip instances, especially in a 2 dof motion where we add 90 degree wrist rotation to the task in the paper. The results show that trajectory adaptation is effective for preventing both translational and rotational slip. Both of these slip types are equally important in this context; however, since we do not have any assumption about the type of slip signal, the idea can be generalised to pure translational slip too.
>
> ` In Fig. 2 it would be easier to compare the rotations if all the y-axes were the same. `
>
> Updated figures with the same y-axis are attached to the comment *Structure of The Responses* on top of the page.
>
> ` is it correct to assume that the red line in (d) and (e) corresponds to the reference trajectory? `
>
> Yes, the red line is the reference trajectory. We added a legend to figures 2 (d) and (e) (Updated figures are attached to the comment *Structure of The Responses* on top of the page).
>
> ` How is the time window C chosen for RSC and PSC? Likewise, how was the prediction horizon h chosen? `
>
> We added details about the planning horizon and context frame lengths chosen in "Experimental results and discussion". The planning horizon is chosen by trading off between classification scores and controller success rate. Longer horizons decrease the classification performance because the slip prediction models should classify slip states far in the future; On the other hand, long horizons would give the proactive controller more reaction time and increases its success rate by having a larger safety margin. By testing horizons of lengths 5, 10, 15, and 20, we picked 10 as the best case. For the context frames length $C$, considering (i) the sensory data frequency (60 hz), and (ii) task completion time (1.2-3 seconds), we chose $C=10$ as a reasonable value including past 0.17 seconds tactile data. The page limit does not allow to include this discussion in the paper but we will add it in the Appendix with corresponding quantitative and qualitative discussion for choosing the horizon and context lengths.
>
> ` The explanation of RSC mentions that formulation was found empirically. Is this referring to the form of the cost function or the constants? `
>
> This refers to both choosing the form of the objective function and the constants. As for the form of the objective function, the most intuitive case is to consider slip signal $S$ being directly multiplied by the velocity norm in equation (1). However, the resulted trajectories were non-smooth and this had a negative effect on slip avoidance. After trying different polynomial and exponential forms, the exponential form presented in equation (1) resulted in the best performance in terms of slip avoidance and trajectory smoothness.
>
> ` Is there any intuition on why PSC performs better with a small (2) set of basis functions? In particular, are there thoughts on why RSC needs more (5) basis functions? `
>
> We discuss the intuition behind the performance of each model with different numbers of basis functions in **General Response 2.2 and 2.3**.

---

> > ### Author Response · Authors · 2022-08-22
> > **Author Response to Reviewer FJ4L, part2**
> >
> > ` Sec 3 mentions that uSkin can be challenging to calibrate. Given that this was mentioned, did this impact the results in any way `
> >
> > The main benefit of calibration is that the tactile readings have force values with units that are interpretable (especially when knowing object weight) as opposed to unitless raw values in our case. It did not affect this method since tactile readings are processed by deep neural networks (DNN) (and standardized prior to that) and the variation in the readings resulted from being not calibrated was learned by the DNN. It can affect approaches that use model-based slip detection methods.
> >
> > ` and are there tricks that can be done to mitigate this issue? `
> >
> > A data-driven calibration with a large data set that includes various contact geometry can be used to map raw sensory output to force values.
> >
> > ` With respect to the training set object, it would be helpful to mention, perhaps in the appendix, what weights were used. Is it also correct to assume that the metal pieces were fixed in the box such that they could not move? `
> >
> > The range of weight can vary between 200-600 grams. The metal pieces are large metal bolts that are fixated inside the box and do not move inside the object during the experiments.
> >
> > ` It might be more beneficial to place the data set section after the methodology section. `
> >
> > We listed the revisions we did in **General Response 3**. We believe these revisions substantially improved the clarity and structure of the paper.
> >
> > ` On the first page, line 34 there is a missing close parent. `
> >
> > The paper is carefully read and corrected multiple times for grammatical mistakes.

---

### Author Response · Authors · 2022-08-22
**General Response 3**

**Manuscript revisions**: We revised the paper according to reviewers' comments which significantly improved the clarity and quality of the paper. These are the list of revisions to the manuscript (updated figures/tables are attached to the comment *Structure of The Responses* on top of the page):

*  **Reviewer b9H8, xX6y, and FJ4L**, We moved Section 4 Methodology before Section 3 Dataset.
*  **Reviewer b9H8**, We divided Methodology to two subsections namely (i) Offline slip classification and (ii) Online trajectory adaptation.
*  **Reviewer b9H8**, Figure 1 (c) is now separated from Figure 1 and moved to the Methodology section.
*  **Reviewer xX6y**, We updated Figure 1 (a) to show object rotation (slip) and the reference Cartesian trajectory in the figure. We updated both Figure 1 (a) and (b) to have larger font size and removed the blurriness (please see updated figures in the attachment).
*  **Reviewer xX6y**,  We added tactile, action, and slip names to Figure 1 (c) (please see updated figures in the attachment).
*  **Reviewer PezY**, We removed customised optimality metrics from Table 1 as it caused readers confusion. Instead, we added two new columns namely tracking performance and number of time steps with $rotation>6^{\circ}$ in task execution time (please see the updated table in the attachment).
    We originally had these two columns combined in customised optimality. These two sets of values allow clearer analysis of the performance of PSC and RSC.
* **Reviewer FJ4L**, We updated Figure 2 (a), (b), and (c) to have same y-axis range for better comparison. Figures 2 (d) and (e) are updated to have legend for the reference trajectory (please see updated figures in the attachment).
* **Reviewer PezY and FJ4L**, In Section 5 Experimental Results and Discussion we added new discussion analysing each methods (RSC and PSC) reference tracking and slip avoidance behaviour based on different numbers of basis functions by updated Table 1.
*  **Reviewer xX6y**, We updated the caption of Figure 2 to include each test objects' name.
*  **Reviewer PezY and FJ4L**, Object weight variation is added in Section 5 Experimental Results and Discussion.
*  **Reviewer b9H8, xX6y, PezY, and FJ4L**, The paper is carefully read and corrected multiple times for grammatical errors.
*  **typos** In the revision typos in lines 23, 34, 38, 44, 69, 96 and 246 are corrected.

---

### Author Response · Authors · 2022-08-22
**General Response 2**

**Slip controllers' performances with different numbers of basis functions**: We realised the customised optimality metric (COM) caused readers' confusion about slip controllers' performances.
We acknowledged this and removed it from the table 1. Instead, we computed and added two scores (where their combination made COM in the original presentation of our paper) (i) distance to reference trajectory and (ii) instances of rotations larger than $6^{\circ}$ in two columns in Table 1 (updated table is attached to the comment *Structure of The Responses* on top of the page). The "Distance to reference" score is the sum of Euclidean distance between the reference and adapted trajectories over the task execution time. The "RTS" (Rotation $>6^{\circ}$ Time Steps) score is the number of time steps that the object had rotations larger than 6 degrees (we empirically selected this value). The larger rotations and large number of cases of RTS indicate higher probability of failure. Nonetheless, for all of the controllers in Table 1 (all rows of the table), the controller were fully successful in avoiding task failure. We analysed the performances of slip prevention controller according to these two metrics:

(**General Response 2.1**) **PSC vs RSC**: we see a smaller distance to reference for RSC relative to PSC for each number of basis functions. While PSC sees a prediction of slip it has enough time to deviate enough from the reference to avoid slip more effectively. This leads to having smaller RTS and larger distance to reference for PSC. On the other hand, RSC has a shorter reaction time relative to PSC, which leads to larger RTS. In terms of improving RTS (slip prevention performance), maximum improvement in RSC by changing the number of basis functions is $\frac{35-28}{35}\times 100 = 20$ \%. While switching from RSC to PSC with 5 basis functions for both, improves slip prevention by $\frac{28-8}{28}\times 100 = 71$ \%. This improvement can be higher for other number of basis functions. Separating tracking and rotation scores demonstrates the advantage of PSC over RSC in a more clear fashion.

(**General Response 2.2**) **RSC**: depending on how many basis functions are used, the overlapping area of the first Gaussian with the next ones can change (since we take only the first element of the generated action sequence in the horizon, the change in the beginning part of the horizon is most important). For instance, for 2 basis functions, only the first one is used for generating the first action element. While for the case of 5 basis functions, the first three have an effect on the first action element generation. This change has positive effect on having smaller RTS for RSC up to 5 basis functions. After this point it can have a negative effect on RTS. For reference tracking, increasing the number of basis functions improves tracking performance.

(**General Response 2.3**) **PSC**: we see 2 basis functions resulted zero RTS with largest distance to the reference. Since the controller has fewer degrees of freedom in this case, the cost it needs to pay for preventing slip is to get much farther from the reference (the generated trajectories can be seen in Figure 2 (a) in the Appendix). This is a very conservative proactive controller and based on how much weight we want to put on slip prevention w.r.t reference tracking, we could prefer this controller. In fact, the number of basis functions is a control parameter that we can use to specify how much conservative we want to be about slip avoidance. The safety margin is achieved by reducing the number of basis functions, but the cost is getting farther from the reference trajectory. Changing the basis functions from 2 to 5 for PSC, improves tracking performance by 6\%, 4\%, 15\% respectively, but going further to 6 basis functions has large negative effect on RTS ($\frac{13-8}{8}\times 100 = 62$ \%). As such we choose 5 basis functions to be the best value for PSC based on this quantitative analysis.

---

### Author Response · Authors · 2022-08-22
**General Response 1**

**Limited set of objects and trajectories in the dataset**: We considered this limitation of our work presented in the paper and have continued our work to appear in an extended journal paper (to be submitted).
In our Journal paper, we have collected a larger data set including (i) more diverse classes of trajectories (trapezoidal, cubic, and quintic velocity profiles as well as Kinesthetic teaching motions) and (ii) larger training object set. Regarding stability analysis, we are running test cases using this larger dataset to quantitatively analyse the stability margin and performance of our proposed slip control framework across both (1) objects from different object classes and (2) different movement trajectories (e.g. different velocities, accelerations, and Cartesian movements). Since the core forward model of physical robot interaction is data-driven, there may not exist an analytical proof of stability. Nonetheless, the current manipulation practice involves many slip cases (e.g. in pick-and-place cases at Amzaon or Ocado) including many grasp-slip-drop-grasp cycles as (1) grip force cannot be increased (e.g., (a) a delicate object may not resist a large grip force or (b) the grip force is already at its maximum value), or (2) the grip force control is not available during manipulative movements with off-the-shelf manipulators, such as Franka Emika Panda arm which we used in this paper. However, this paper aims at reducing the number of slip cases available in current practices,e.g., at Amazon and Ocado. Although we do not aim to fully guarantee slip prevention, our novel approach helps reduce the number of grasp-slip-drop-grasp cycles and increases the efficiency of the current manipulation process.

---

### Author Response · Authors · 2022-08-22
**Structure of The Responses**

**Comment:**

**Our responses organisation**:
1. General Responses (addressing concerns shared by all reviewers.)
2.  Response to Meta Review
3. Responses to each reviewer.

Updated Figures and Tables are **attached** to this comment.

**Zip File:**

/attachment/7ac98a2d8b58abe97453192b59b0e2e321139c9d.zip

---

### Meta-Review · Area_Chair_nXbT · 2022-08-10

**Recommendation:** Accept (Poster)
**Confidence:** 5

**Metareview:**

This paper presents a method for preventing slips in point-to-point motions while grasping an object using LSTM-based models for detecting and predicting slips. This approach is effective when contact force cannot be increased. Reactive and proactive controllers are presented, and the latter demonstrated better performance.

The paper proposes an interesting approach to slip prevention especially useful for handling delicate objects. The authors successfully addressed most of the concerns raised by the reviewers, while expanding the dataset and examples will be addressed in a future journal paper. Nevertheless, the paper will make a valuable contribution to the conference.

**Best Paper Nomination:**

No

---

> ### Author Response · Authors · 2022-08-22
> **Author Response to Area Chair**
>
> Dear Area Chair, we thank you for your technical comments about the strengths and weaknesses of the paper.
>
> ` Limited task/motion variation in the dataset `
>
> We aimed to demonstrate a proof of concept of using trajectory adaptation instead of grip force control for slip avoidance. As such, the main variation we considered to have in the dataset was different acceleration/deceleration values to be able to analyse the performance of the proposed novel slip controllers in a semi-structured task. A detailed description of the extension of this paper is described in **General Response 1** where we explain how we addressed task/motion variation limitations. Including the dataset and experiments described in **General Response 1** in the paper is not within the context of the contributions of the paper considering the page limitation.
>
> ` Performance drop with more basis functions `
>
> In **General Response 2** we clarified the analysis of the performance of each slip controller (RSC and PSC) with different numbers of basis functions. Based on the presented discussion, the number of basis functions can be used as a safety parameter to tune how much conservative the controller can be about slip prevention and reference tracking; Also showing the performance does not drop with more basis functions. In the extension work described in **General Response 1**, we relaxed the 1 dof motion constraint and execute the optimization for all 6 dof in real-time. We tested the model with different classes of reference trajectories (polynomials of different orders and trapezoidal) with novel start and end points. The results show with the new dataset, the proposed trajectory adaption can deal with more complex tasks and still do effective slip prevention.